# Measurement-Induced Entanglement Phase Transition in Random Bilocal Circuits

Xuyang Yu[1] and Xiao-Liang Qi[2]

[1]*Department of Physics, University of California, Berkeley, California 94720 USA*
[2]*Stanford Institute for Theoretical Physics, Stanford University, Stanford, California 94305, USA*
(Dated: February 2, 2022)

Measurement-induced entanglement phase transitions, caused by the competition between entangling unitary dynamics and disentangling projective measurements, have been studied in various random circuit models in recent years. In this paper, we study the dynamics of averaged purity for a simple $N$-qudit Brownian circuit model with all-to-all random interaction and measurements. In the large-$N$ limit, our model is mapped to a one-dimensional quantum chain in the semi-classical limit, which allows us to analytically study critical behaviors and various other properties of the model. We show that there are two phases distinguished by the behavior of the total system entropy in the long time. In addition, the two phases also have distinct subsystem entropy behavior. The low measurement rate phase has a first-derivative discontinuity in the behavior of second Renyi entropy versus subsystem size, similar to the "Page curve" of a random state, while the other phase has a smooth entropy curve.

**Introduction**. In recent years, a lot of new progress has been made in understanding quantum entanglement in many-body physics. In general, time evolution of a many-qubit system introduces entanglement between different qubits. In the long time, entanglement entropy of a subsystem tends to approach its maximum, or maximal value allowed by symmetry constraints, which is known as the phenomenon of thermalization. If we apply quantum measurements to single-qubit (or few-qubit) operators, the measurement can project the many-body wavefunction to a less entangled state. Consequently, if the measurement occurs with a finite rate, it competes with the thermalization and entanglement growth, which leads to the interesting phenomenon of measurement-induced phase transition (MIPT)[1–4]. MIPT has been studied in various models of quantum dynamics, such as random circuits and random Clifford circuits.[5–11] In models with spatial locality, MIPT is usually a phase transition between a volume law entropy phase and an area law phase. In models without spatial locality, the difference between volume law and area law is not well-defined. MIPT in certain nonlocal random circuit models[12–16] and the Brownian SYK model[16, 17] have been studied. However, the results on non-local models have mainly focused on the long-time entropy of an mixed initial state (known as the purification phase transition), and they rely on phenomenological effective field theory and/or numerical results.

In this paper, we propose a simple model in which various properties related to MIPT can be studied analytically for the second Renyi entropy. We consider a random bilocal circuit of $N$ qudits with no spatial locality[18, 19], as is illustrated in Fig. 1. Two qudits are randomly chosen and coupled by a two-qudit gate with a certain probability. Quantum measurement is randomly applied to one of these qudits with a certain rate. In the sense of random average, this model has a permutation symmetry between different qudits, which means the purity of a subsystem is only a function of the subsystem size, denoted by $\mathcal{P}_n$, $n = 0, 1, ..., N$. We take a continuous-time limit of this model and show that the time evolution of subsystem purity is described by a linear differential equation. In the large-$N$ limit, we show that MIPT occurs in the average purity of this simple model between a "cusp phase" where the entropy as a function of subsystem size has a first derivative discontinuity at half system size and a smooth phase where the entropy curve is smooth. We show that the purity differential equation can be mapped to a single-particle tight-binding model problem, where the two phases simply correspond to the particle staying in a double-well potential vs. a single-well one. $\frac{1}{N}$ plays the role of $\hbar$ in this single-particle problem. In large-$N$ limit, there are two degenerate ground states in the cusp phase, which is responsible for the finite residual entropy in the long time (if the initial state is a mixed state), and the cusp-shape entropy curve. We obtain various critical behavior of this MIPT analytically. Our model is similar to that of Ref. [15], but our approach allows us to study the averaged purity of this model more directly and analytically.

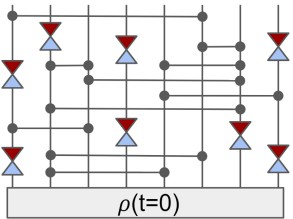

FIG. 1: Illustration of the Brownian circuit model with measurements. The unitary time evolution is generated by random bilocal Hermitian operators that are uncorrelated between different times. At random spacetime locations, one of the qubits is projected to a random state $|\varphi\rangle$, and then replaced by another independently chosen random state $|\psi\rangle$.

**Purity evolution.** The model we study is a Brownian

circuit[18] with $N$ qudits each with a Hilbert space dimension $d$. The time evolution of this system is described by a Hamiltonian that contains random bi-local coupling terms with no correlation between different times:

$$H(t) = \sum_{i<j} J_{ij}^{ab}(t) T_{ia} T_{jb} \qquad (1)$$

$$\overline{J_{ij}^{ab}(t) J_{kl}^{cd}(t')} = \frac{J}{4d^3N} \delta^{kl}_{ij} \delta^c_a \delta^d_b \delta(t - t') \qquad (2)$$

Here $T_{ia}$, $a = 0, 1, ..., d^2 - 1$ is a complete basis of Hermitian operators satisfying the orthonormal condition $\text{tr}\,(T_{ia} T_{ib}) = d\delta_{ab}$. In addition, a measurement occurs at a constant rate. After each short time $\delta t$, there is a small probability $p = N(d+1)\lambda\delta t$ that one of the qudits, randomly chosen, is measured. The measured qudit is projected by applying the operator $|\psi\rangle\langle\varphi|$ with $|\psi\rangle$ and $|\varphi\rangle$ independently random states. It should be noted that after the projective measurement on $|\varphi\rangle$, a different random state $|\psi\rangle$ is created. Physically, this can be viewed as a projective measurement followed by a random single site unitary $U_1 |\varphi\rangle\langle\varphi| = |\psi\rangle\langle\varphi|$. Since the dynamics are described by a random bilocal Hamiltonian, the ensemble of which is invariant under (conjugation of) random unitary, adding this additional single-site unitary will not change our discussion, but it simplifies the derivation.

This hybrid Brownian evolution leads to a trajectory of (normalized) states $\rho(t)$ which is a function of random parameters $J_{ij}^{ab}(t)$, location of the measurement events $i_s, t_s$ and random states $|\varphi(t_s)\rangle_{i_s}, |\psi(t_s)\rangle_{i_s}$. For simplicity we denote all these parameters together as $\zeta$, which is associated with a probability $p_\zeta(t)$. We will study the following averaged purity of a generic subsystem $Q$:

$$e^{-S_Q^{(2)}} \equiv \frac{\mathcal{P}_Q}{\mathcal{P}_\emptyset} \equiv \frac{\int d\zeta\, p_\zeta(t)^2 \text{tr}_Q\left[\rho_Q^2(t)\right]}{\int d\zeta\, p_\zeta(t)^2} \qquad (3)$$

with purity $\mathcal{P}_Q$ defined by the numerator. Physically, after we carry out the measurement on two copies of the circuit at the same spacetime location, we average the purity over the resulting states conditioned on identical measurement results in these two copies. In the supplemental material [20], we discuss the relation of this average to the averaged von Neumann entropy.

Note that the unnormalized state $\sigma(t) \equiv p_\zeta\rho(t)$ is linear in the initial state $\rho(0)$, which allows us to derive a simple master equation for $\mathcal{P}_Q$. We will sketch the idea of the derivation here, with details reserved to the supplemental materials[20]. In the Hilbert space of two copies of the $N$-qudit system, we can define $X_Q$ as the swap operator that permutes the two replicas in the $Q$ region and keeps the complement unaffected. It is well-known that $\text{tr}_Q\left[\sigma_Q^2(t)\right] = \text{tr}\left[X_Q\sigma(t)\otimes\sigma(t)\right]$ is the expectation value of $X_Q$ operator in the two-replica state $\sigma(t)\otimes\sigma(t)$. The time evolution from 0 to $t$ can be written as

$$V_\zeta(t) = T\left[e^{-i\int dt H(t)} \prod_s |\psi(t_s)\rangle_{i_s} \langle\varphi(t_s)|_{i_s}\right] \qquad (4)$$

with $T$ stands for time-ordering. Then

$$\mathcal{P}_Q = \int d\zeta\, \text{tr}\left[X_Q V_\zeta(t)^{\otimes 2} \sigma(0)^{\otimes 2} V_\zeta^\dagger(t)^{\otimes 2}\right]$$
$$\equiv \text{tr}\left[\overline{\hat{X}_Q(t)}\sigma(0)^{\otimes 2}\right] \qquad (5)$$

Here $\overline{\hat{X}_Q(t)}$ is the random average of Heisenberg operator $\hat{X}_Q(t) = V_\zeta^\dagger(t)^{\otimes 2} X_Q V_\zeta(t)^{\otimes 2}$. Without averaging, the evolution of $\hat{X}_Q$ will depend on many other operators in the doubled system. However, after averaging over random parameters, one can show that $\hat{X}_Q(t)$ evolves to a linear superposition of swap operators in different regions, which leads to a differential equation[20]

$$\frac{d}{dt}\overline{\hat{X}_Q(t)} = J\sum_R M_{QR}\overline{\hat{X}_R(t)} \qquad (6)$$

where $R$ runs over all sub-regions of the system. For later convenience, we define the matrix with a prefactor of $J$ such that $M_{QR}$ is dimensionless. $M_{QR}$ only has nontrivial matrix elements when $R = Q$, or when $R$ and $Q$ are different by removing one site and/or adding one site. Taking expectation value in the initial state $\sigma(0)^{\otimes 2}$, we obtain the same differential equation for $\mathcal{P}_Q$: $\dot{\mathcal{P}}_Q = J\sum_R M_{QR}\mathcal{P}_R$. In addition, after averaging there is a permutation symmetry between different qudits so that $\mathcal{P}_Q$ only depends on the size of subsystem $Q$. We can denote $\mathcal{P}_Q = \mathcal{P}_n$ when the size of $Q$ is $|Q| = n$, $n = 0, 1, ..., N$. This further simplifies the differential equation into that of the $(N+1)$-dimensional vector $\mathcal{P}_n$:

$$J^{-1}\dot{\mathcal{P}}_n = \sum_{m=0}^N M_{nm}\mathcal{P}_m$$
$$\equiv a_n\mathcal{P}_n + b_n\mathcal{P}_{n+1} + c_{n-1}\mathcal{P}_{n-1} \qquad (7)$$

with $a_n = -\left(d + \frac{1}{d}\right)\frac{(N-n)n}{N} - \alpha Nd\left(d + 1 - \frac{1}{d}\right)$,

$b_n = \frac{(N-n)n}{N} + \alpha(N-n)$, $c_{n-1} = \frac{(N-n)n}{N} + \alpha n$

where we have defined a dimensionless variable $\alpha = \frac{\lambda}{dJ}$ as ratio of the rate of measurement to the rate of unitary evolution, and we have also set $J = 1$ for all numerical calculations. Eq. (7) plays a central role in this paper, which (in the large-$N$ limit) determines the MIPT and critical exponents that will be discussed below. Eq. (7) for $\alpha = 0$ describes a unitary random Brownian circuit, which was first proposed in Ref. [18]. A generalization to Brownian circuit coupled with bath has been studied in Ref. [19].

Before presenting analytic results in the large-$N$ limit, we would like to first show some numerical evidence of the MIPT. We first consider an initial state which is maximally mixed on one site and pure state on other sites, $\sigma(0) = \frac{1}{d}\mathbb{I}_i \otimes_{j\neq i} |0\rangle\langle 0|$. Then averaging over the site $i$,

we have $\mathcal{P}_n(0) = \frac{N-n}{N} + \frac{n}{N}\frac{1}{d}$. Then we solve the differential equation (7) with this initial condition. In the time region $N \ll tJ \ll e^N$, $\mathcal{P}_n(t \to \infty)$ decays exponentially, but the ratio $e^{-S_n^{(2)}} = \frac{\mathcal{P}_n}{\mathcal{P}_0}$ saturates to a constant. Fig. 2 (a) shows the second Renyi entropy of the entire system in the long-time limit $S_N^{(2)}(t \to +\infty)$ as a function of $\alpha$. We see that the entropy is finite for $\alpha < 0.5$ and vanishes for $\alpha > 0.5$, suggesting a phase transition at $\alpha = 0.5$. (The numerics are carried for $d = 2$.) This transition is called the purification phase transition in [14]. As another probe of the phase transition, we study the case that the initial state is a pure state with $\mathcal{P}_n(0) = 1$. Fig. 2 (b) shows the entropy $\frac{1}{N}S_n^{(2)}(t \to +\infty)$ as a function of $n/N$ for different $\alpha$. We observe that for $\alpha < 0.5$ the entropy curve has a cusp (discontinuity of the first derivative) at $\frac{n}{N} = \frac{1}{2}$. For $\alpha > 0.5$ the entropy is a smooth function of $\frac{n}{N}$. This further supports the observation that there is a phase transition at $\alpha = 0.5$. In the following, we will study the large-$N$ limit of Eq. (7) analytically, which confirms the numerical observation and provides a comprehensive understanding of MIPT in this model.

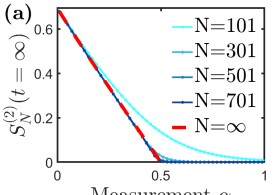
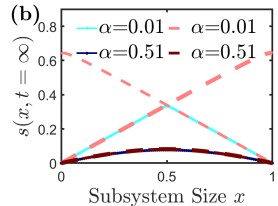

FIG. 2: (a) Numerical and analytical results for the long-time second Renyi entropy of the entire system when the initial state contains a single maximally-entangled qubit. The calculation is done for $d = 2$ and the transition occurs at $\alpha_c = 0.5$. (b) The subsystem second Renyi entropy density $s(x) = S_n^{(2)}/N$ versus $x = n/N$, for $\alpha < \alpha_c$ and $\alpha > \alpha_c$ (solid lines). The dashed lines are analytic results on the contribution by different saddle points (see text).

**Large-$N$ limit**. The time evolution of $\mathcal{P}_n$ is determined by the eigenvalues and eigenvectors of matrix $M$ in Eq. (7). $M$ is non-Hermitian, but as a tridiagonal matrix, it can be transformed to a Hermitian matrix by a similarity transformation

$$\mathcal{P}_n = \Lambda_n \phi_n, \quad -JM_{nm}\Lambda_m\Lambda_n^{-1} = H_{nm}, \quad (8)$$

$$\text{with } \Lambda_0 \equiv 1, \ \Lambda_{n>0} \equiv \prod_{m=1}^{n}\sqrt{\frac{c_{m-1}}{b_{m-1}}} \quad (9)$$

$H$ is a symmetric tridiagonal matrix with

$$H_{nn} = -Ja_n, \ H_{n-1,n} = -N\tau_n \equiv -J\sqrt{b_{n-1}c_{n-1}} \quad (10)$$

$J^{-1}H$ has the same eigenvalue as $-M$. The vector $\phi_n$ satisfy the differential equation $\dot{\phi}_n = -\sum_m H_{nm}\phi_m$,

which means $\phi_n$ behaves as a wavefunction under imaginary time evolution with Hamiltonian $H$. The Hamiltonian describes a single particle hopping on a one-dimensional lattice, where both the hopping term and the on-site potential are position-dependent. If we denote the eigenvalues of $H$ as $E_a$ and eigenvectors as $\phi_{a,n}$, $a = 0, 1, 2, ..., n$ (with $E_a \leq E_{a+1}$), then the time evolution of purity is given by

$$\mathcal{P}_n(t) = \Lambda_n \sum_a \eta_a \phi_{a,n} e^{-E_a t} \quad (11)$$

$$\eta_a = \sum_n \phi_{a,n}^* \mathcal{P}_n(0)\Lambda_n^{-1} \quad (12)$$

where every term without an explicit $t$ is time-independent. In the limit $t \to \infty$, $\mathcal{P}_n(t)$ is determined by the lowest energy eigenvector. If the lowest energy eigenvalue is unique, $\mathcal{P}_n(t)/\mathcal{P}_0(t)$ in the limit $t \to \infty$ will be proportional to $\Lambda_n \phi_{0,n}$ regardless of the initial state. In Fig. 2 we observe that in the small $\alpha$ phase the final state entropy $S_N^{(2)}(\infty)$ depends on the initial state, which suggests that $H$ has a ground state degeneracy. Indeed, we can directly verify that $E_1 - E_0$ vanishes for large $N$ for $\alpha < 0.5$, as is shown in Fig. 3 (a).

To understand this ground state degeneracy of $H$, we consider a continuous limit of the eigenvalue equation $\sum_m H_{nm}\phi_m = E\phi_n$. The numerics suggest that $\frac{1}{N}S_n^{(2)}$ is finite in large-$N$ limit for fixed $x = \frac{n}{N}$. Since $\log \Lambda_n$ is also $\propto N$ (see below), so is $\log \phi_n$. Thus we can take the ansatz

$$\phi_n = e^{-NA(x)}, \ x = \frac{n}{N} \quad (13)$$

Assuming $A(x)$ to be a smooth function of $x$, the eigenvalue equation has the following continuous limit:

$$\epsilon \equiv \frac{E}{N} = V(x) - 4\tau(x)\sinh^2\left(\frac{1}{2}\partial_x A(x)\right) \quad (14)$$

$$\text{with } V(x) = -Ja_n/N - 2\tau_n, \ \tau(x) = \tau_n \quad (15)$$

The second Renyi entropy per qudit is determined by $A(x)$:

$$s(x) = \frac{1}{N}S_n^{(2)} = A(x) + D(x) - A(0) \quad (16)$$

with $D(x) = -\frac{1}{N}\log \Lambda_n = -\int_0^x dx' \log \sqrt{\frac{x'(1-x')+\alpha x'}{x'(1-x')+\alpha(1-x')}}$. The ansatz (13) can be view as a WKB approximation for the quantum Hamiltonian $\hat{h} \equiv H/N$ in the classically forbidden region, with $N$ playing the role of $\frac{1}{\hbar}$. Eq. (14) is the analog of Hamilton-Jacobi equation for the classical action $A(x)$, with momentum $p \equiv \partial_x A$. The kinetic energy term is proportional to $\sin^2 \frac{p}{2}$ rather than $p^2$, which is consistent with the fact that the original model is a tight-binding model.

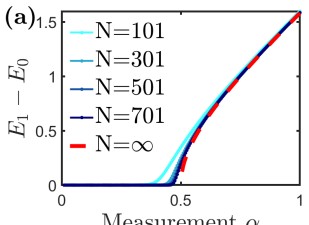 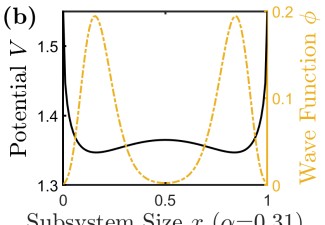 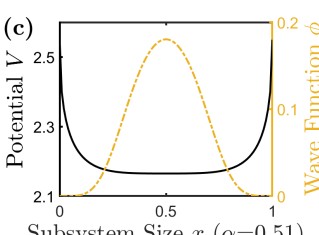 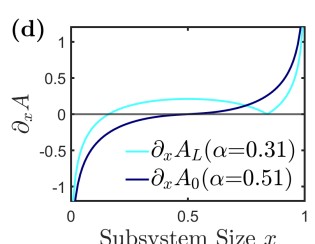

FIG. 3: (a) Difference of the two lowest energies of the Hamiltonian defined in Eq. (8) and (10) for different $N$. For comparison, the red dashed line is the analytic result for $N \to \infty$. (b) and (c) The potential $V(x)$ (Eq. (15)) and symmetric ground-state wavefunction for the two phases, respectively. (d) The blue curve is $\partial_x A_L(x)$ vs. $x$ in the small $\alpha$ phase, with $\phi_L(x) = e^{-N A_L(x)}$ the ground-state wavepacket peaked in the left minimum of $V(x)$. The black curve is $\partial_x A_0(x)$ vs. $x$ in the large-$\alpha$ phase, corresponding to the unique ground-state wavepacket centered at $x = 1/2$.

In this mapping, the large-$N$ limit maps to the classical limit, so that the particle will stay at the minimum of $V(x)$. We see (Fig. 3) that $V(x)$ has two minima for $\alpha < \alpha_c$ and a single minimum for $\alpha \geq \alpha_c$, where

$$\alpha_c = \frac{d-1}{2} \qquad (17)$$

The doubly degenerate ground states for $\alpha < \alpha_c$ correspond to two wavepackets that concentrate in the two minima of $V(x)$, with width $\propto N^{-1/2}$. The ground-state energy is $\epsilon_0 \simeq \min_x V(x) + O\left(\frac{1}{N}\right)$.

**Critical behavior.** The behavior of second Renyi entropy in the long time is completely determined by the ground-state wavefunction(s), which can be solved by an integration over $\partial_x A$ obtained from Eq. (14) (in which we take $\epsilon = \epsilon_0 \equiv \min_x V(x)$ to the leading order in large $N$). There are two roots of $\partial_x A$, which we need to choose by continuity condition. Fig. 3 (d) illustrates the choice of $\partial_x A$ for both phases. In the cusp phase, there are two choices symmetric according to $x = 1/2$, which correspond to the two ground-state wavepackets, denoted as $\phi_L(x) = e^{-N A_L(x)}$ and $\phi_R(x) = e^{-N A_R(x)} = \phi_L(1-x)$. For $\alpha > \alpha_c$, there is a unique wavepacket centered at $x = 1/2$, which we can denote as $\phi_0(x) = e^{-N A_0(x)}$.

The ground-state wavefunction(s) completely determine the long-time behavior of $\mathcal{P}_n$ according to Eq. (11) and (12). In the following we will analyze the behavior of $s(x)$ and extract relevant critical exponents. For $t \to \infty$, for $\alpha < \alpha_c$ we have

$$-\log \frac{\mathcal{P}_n(t \to \infty)}{\mathcal{P}_0(t \to \infty)} = N D(x) - \log \frac{\eta_L \phi_L(x) + \eta_R \phi_R(x)}{\eta_L \phi_L(0) + \eta_R \phi_R(0)} \qquad (18)$$

The coefficients can be expressed as an integral (with an unimportant prefactor that we omitted)

$$\eta_L = \int_0^1 dx \mathcal{P}(x,0) e^{-N[A_L(x)-D(x)]} \qquad (19)$$

In the large-$N$ limit, this integral (assuming $\mathcal{P}(x,0)$ is a smooth function, as is in the case of an order-1

initial state entropy) is dominated by the minimum of $A_L(x) - D(x)$. Approximately, $\eta_L \simeq \sqrt{2\pi/(N k_L)} \mathcal{P}(x_L, 0) \exp\{-N \min_x [A_L(x) - D(x)]\}$ with $x_L$ the minimum location of $A_L(x) - D(x)$, and $k_L$ the second derivative at the minimum. Similarly we found $\eta_R$ determined by the saddle point $x_R = 1 - x_L$. Eq. (18) in the large-$N$ limit leads to

$$S^{(2)}(x=1, t \to \infty) = -\log \frac{\eta_R \phi_R(1)}{\eta_L \phi_L(0)} = -\log \frac{\mathcal{P}(x_R,0)}{\mathcal{P}(x_L,0)}$$
$$= S^{(2)}(x_R,0) - S^{(2)}(x_L,0) \qquad (20)$$

Here we have used the fact that $\phi_L(1)/\phi_R(1) = \phi_R(0)/\phi_L(0) = e^{-N[A_L(1)-A_R(1)]}$ is exponentially small. Solving $x_L$ analytically leads to

$$x_L - \frac{1}{2} \simeq \frac{1}{d-1}(\alpha - \alpha_c) \qquad (21)$$

$$S^{(2)}(x=1, t \to \infty) \simeq \frac{4}{d+1}(\alpha_c - \alpha) \qquad (22)$$

near the critical point, which is consistent with the numerics.

It should be noted that for a finite $N$ system this long-time entropy will eventually vanish in a much longer time scale $\sim e^N$. This effect comes from the small splitting between the symmetric and antisymmetric combination of the two ground-state wavepackets, which is of order $e^{-\gamma N}$ with $\gamma$ an order 1 constant. In time $t \gg J^{-1} e^{\gamma N}$, $\mathcal{P}_n(t)$ will be determined by the symmetric wavefunction which corresponds to $S^{(2)}(x=1) = 0$. This is consistent with results in the literature [14].

Eq. (18) also determines the behavior of subsystem entropy. If the initially state is a pure state, there is a symmetry $\mathcal{P}_n = \mathcal{P}_{N-n}$, which in the $\alpha < \alpha_c$ phase requires $\eta_L = \eta_R$. In large-$N$ we obtain

$$s(x, \infty) \equiv -\frac{1}{N} \log \frac{\mathcal{P}_n(t \to \infty)}{\mathcal{P}_0(t \to \infty)}$$
$$= D(x) + \min\{A_L(x), A_R(x)\} \qquad (23)$$

which agrees well with the finite $N$ numerics (Fig. 2 (b)). The switch between $A_L$ and $A_R$ occurs at $x = \frac{1}{2}$, which

is the reason for the discontinuity in the first derivative. Quantitatively, we obtain

$$\partial_x s(x,\infty)|_{x\to\frac{1}{2}^-} = -\log\left(1 + \frac{2}{d}\left(\alpha - \alpha_c\right)\right) \qquad (24)$$

Near $\alpha = \alpha_c$ we obtain $\partial_x s(x,\infty)|_{x\to\frac{1}{2}^-} \simeq \frac{2}{d}\left(\alpha_c - \alpha\right)$.

In addition to the behavior of the long-time limit, we can also study the time evolution of $s(x,t) = -\frac{1}{N}\log\frac{\mathcal{P}_n(t)}{\mathcal{P}_0(t)}$. If we take an initial state $s(x,0) = 0$, we observe that the cusp in the entropy curve actually appears at a finite time. To see that, we can define $\mathcal{P}_n(t) = \Lambda_n\phi_n(t)$ and $\phi_n(t) = e^{-NA(x,t)}$. $A(x,t)$ satisfies the time-dependent version of the Hamilton-Jacobi equation (14):

$$\partial_t A(x,t) = V(x) - 4\tau(x)\sinh^2\left(\frac{\partial_x A(x,t)}{2}\right) \qquad (25)$$

$A(x,t)$ determines the entropy per qudit $s(x,t) = A(x,t) + D(x)$. When the initial state is a pure state, the solution has the symmetry $A(x,t) = A(1-x,t)$. Assuming $A(x,t)$ to be smooth for early time, we have $\partial_x s(x)|_{x=1/2} = 0$. Define $u(t) = -\partial_x^2 s(x)|_{x=1/2}$, we can derive an equation of $u(t)$ from Eq. (25):

$$J^{-1}\dot{u} = \tilde{a}\left(u - \tilde{u}\right)^2 + \tilde{b} \qquad (26)$$
$$\tilde{a} = \alpha + \frac{1}{2}, \ \tilde{b} = \frac{2d^2 + 2}{d} - \frac{2(1 + 2\alpha)^2 + 2}{1 + 2\alpha}$$
$$\tilde{u} = -\partial_x^2 D(x)|_{x=1/2}$$

For $\alpha > \alpha_c$, $u(t)$ saturates to a finite value in $t \to \infty$. For $\alpha < \alpha_c$, $u(t)$ diverges at a finite time $t_c$. Near the phase transition,

$$t_c = t_0 + \frac{\pi}{2J\sqrt{\tilde{a}\tilde{b}}} \simeq \frac{\pi}{J\sqrt{2\left(d - \frac{1}{d}\right)\left(\alpha_c - \alpha\right)}} \qquad (27)$$

**Discussion.** In summary, we have studied a simple exactly solvable model that describes a MIPT of averaged purity. A natural question is whether a similar transition occurs if we average over von Neumann entropy or Renyi entropy, such as $\int \zeta p_\zeta S_Q$ with $S_Q$ the von Neumann entropy. Using Eq. (6), the purity differential equation can also be generalized to systems with locality. Another question is whether the differential equation approach can be applied to study other physical problems in quantum chaos, such as operator size growth. It is also interesting to study the relation between the purity differential equation and emergent spacetime geometry. In holographic duality, the cusp phase and smooth phase for subsystem entropy correspond to a geometry with and without an "entanglement shadow"[21, 22], respectively. It will be interesting to explore whether there are comparisons that can be made about critical behaviors of the transition between these phases.

**Acknowledgment.** We would like to thank Yimu Bao for helpful discussions. This paper is supported by the National Science Foundation under grant No. 2111998, and the Simons Fundation. This work is also supported in part by the DOE Office of Science, Office of High Energy Physics, the grant de-sc0019380. This work is partially finished when XLQ is visiting the Institute for Advanced Study, Tsinghua University (IASTU). XLQ would like to thank IASTU for hospitality.

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

arXiv:2111.05355 (2021).

# SUPPLEMENTARY MATERIALS

## Contents

### Derivations of the Purity Differential Equation

From Eq. (4) to Eq. (7): The main results of our work depend on the purity differential equation Eq. (7). Here we derive the differential equation using the definition of the Hamiltonian in Eq. (1) and the definition of measurements in Eq. (4). We copy them here for convenience.

$$H(t) = \sum_{i<j} J_{ij}^{ab}(t) T_{ia} T_{jb}, \quad \overline{J_{ij}^{ab}(t) J_{kl}^{cd}(t')} = \frac{J}{4d^3 N} \delta_{ij}^{kl} \delta_a^c \delta_b^d \delta(t - t') \tag{S.1}$$

and $\hat{X}_Q(t + \Delta t) = V_\zeta(\Delta t) \hat{X}_Q(t) V_\zeta^\dagger(\Delta t)$ with the non-unitary operator

$$V_\zeta(\Delta t) = T \left[ e^{-i \int_0^{\Delta t} dt H(t)} \prod_{s(\Delta t)} |\psi(t_s)\rangle_{i_s} \langle\varphi(t_s)|_{i_s} \right] \simeq V_U(\Delta t) V_M(\Delta t) \tag{S.2}$$

$$V_U(\Delta t) = T \left[ e^{-i \int_0^{\Delta t} dt H(t)} \right] \tag{S.3}$$

$$V_M(\Delta t) = T \prod_{s(\Delta t)} |\psi(t_s)\rangle_{i_s} \langle\varphi(t_s)|_{i_s} \tag{S.4}$$

with $T$ stands for time-ordering.

Recall that in Eq. (5) we have

$$\mathcal{P}_Q = \int D\zeta \, \mathrm{tr} \left[ X_Q V_\zeta(t)^{\otimes 2} \sigma(0)^{\otimes 2} V_\zeta^\dagger(t)^{\otimes 2} \right] \equiv \mathrm{tr} \left[ \overline{\hat{X}_Q(t)} \sigma(0)^{\otimes 2} \right] = \mathrm{tr} \left[ X_Q \overline{\sigma(t)}^{\otimes 2} \right] \tag{S.5}$$

where the overline denotes average over $\zeta$. We aim to write down an expression for $\dot{\mathcal{P}}_Q$ and we use the last expression above for clarity. The derivation below is similar to that in Ref. [19]. We first use the Choi-Jamiolkowski mapping which allows us to interpret the operators defined on two replicas $\mathcal{H} \otimes \mathcal{H}$ as a state in four replicas $\mathcal{H}^{(1)} \otimes \mathcal{H}^{(1)} \otimes \mathcal{H}^{(2)} \otimes \mathcal{H}^{(2)}$. We have also written $V_\zeta(\Delta t)$ as $V_\zeta$ for brevity.

$$\overline{|\sigma(t + \Delta t)^{\otimes 2}\rangle\rangle} \equiv \overline{(\mathbb{1}_\mathcal{H} \otimes V_\zeta \sigma(t) V_\zeta^\dagger \otimes \mathbb{1}_\mathcal{H} \otimes V_\zeta \sigma(t) V_\zeta^\dagger)} |I^+\rangle_{1,\dots,N}$$

$$= \overline{V_\zeta^* \otimes V_\zeta \otimes V_\zeta^* \otimes V_\zeta} \, \overline{|\sigma(t)^{\otimes 2}\rangle\rangle} \tag{S.6}$$

$$\langle\langle X_Q| \equiv \bigotimes_{i \in Q} \langle I_i^-| \bigotimes_{j \in \bar{Q}} \langle I_j^+| \tag{S.7}$$

where we introduced the maximally entangled state $|I^+\rangle_{1,\dots,N} = \bigotimes_{i=1}^N |I_i^+\rangle$ and swap state $\langle\langle X_Q|$, with definitions

$$|I_i^+\rangle = \sum_{a,b=0}^{d-1} (|a\rangle_i \otimes |a\rangle_i) \otimes (|b\rangle_i \otimes |b\rangle_i), \quad |I_i^-\rangle = \sum_{a,b=0}^{d-1} (|a\rangle_i \otimes |b\rangle_i) \otimes (|b\rangle_i \otimes |a\rangle_i) \tag{S.8}$$

where it is evident that $X_Q$ permutes the two replicas in the $Q$ region. So now

$$\dot{\mathcal{P}}_Q = \langle\langle X_Q| \frac{d}{dt} \overline{|\sigma(t) \otimes \sigma(t)\rangle\rangle} = \lim_{\Delta t \to 0} \frac{1}{\Delta t} \langle\langle X_Q| \left( \overline{V_\zeta^* \otimes V_\zeta \otimes V_\zeta^* \otimes V_\zeta} - \mathbb{I} \right) \overline{|\sigma(t) \otimes \sigma(t)\rangle\rangle}$$

$$\equiv \langle\langle X_Q| (-\mathcal{L}) \overline{|\sigma(t) \otimes \sigma(t)\rangle\rangle} \equiv \langle\langle X_Q| (-\mathcal{L}_U - \mathcal{L}_M) \overline{|\sigma(t) \otimes \sigma(t)\rangle\rangle} \tag{S.9}$$

in terms of $\mathcal{P}$. Recall that $V_\zeta = V_U V_M$. For clarity, we plan to calculate the random-averaged Linbladians $\mathcal{L}_U$ and $\mathcal{L}_M$ individually and act each of them on the bra state $\langle\langle X_n|$. First, we derive the Linbladian $\mathcal{L}_U$ corresponding to the unitary evolution after the disorder averaging. For a short time $\Delta t$, we have

$$\overline{V_U^* \otimes V_U \otimes V_U^* \otimes V_U} = \overline{\exp(iH^*\Delta t) \otimes \exp(-iH\Delta t) \otimes \exp(iH^*\Delta t) \otimes \exp(-iH\Delta t)} \tag{S.10}$$

where $\exp(-iH\Delta t)$ is a shorthand for

$$\exp\left(-i\int_t^{t+\Delta t} H(t')\, dt'\right) = 1 - i\int_t^{t+\Delta t} H(t')\, dt' - \frac{1}{2}\left(\int_t^{t+\Delta t} H(t')\, dt'\right)\left(\int_t^{t+\Delta t} H(t')\, dt'\right) + O(\Delta t^3) \tag{S.11}$$

In the four tensor products, the zeroth order of $\Delta t$ is 1, the first order terms canceled with each other, and the second order term is what we really want. Use the definition of $H$ in Eq. (S.1)

$$\overline{\exp(iH^*\Delta t) \otimes \exp(-iH\Delta t) \otimes \exp(iH^*\Delta t) \otimes \exp(-iH\Delta t)} \simeq 1-$$

$$\int_t^{t+\Delta t} dt' \int_t^{t'} dt'' \sum_{\substack{i<j,ab \\ k<l,cd}} \overline{J_{ij}^{ab}(t')J_{kl}^{cd}(t'')} \left(T_{ia}^1 T_{jb}^1 + T_{ia}^2 T_{jb}^2 - T_{ia}^{\bar1*}T_{jb}^{\bar1*} - T_{ia}^{\bar2*}T_{jb}^{\bar2*}\right)\left(T_{kc}^1 T_{ld}^1 + T_{kc}^2 T_{ld}^2 - T_{kc}^{\bar1*}T_{ld}^{\bar1*} - T_{kc}^{\bar2*}T_{ld}^{\bar2*}\right)$$

$$= 1 - \frac{J}{4d^3 N}\Delta t \sum_{i<j,ab}\left(T_{ia}^1 T_{jb}^1 + T_{ia}^2 T_{jb}^2 - T_{ia}^{\bar1*}T_{jb}^{\bar1*} - T_{ia}^{\bar2*}T_{jb}^{\bar2*}\right)^2 \tag{S.12}$$

So the Linbladian for the unitary part is

$$\mathcal{L}_U = \lim_{\Delta t\to 0}\frac{1}{\Delta t}\left[1 - \overline{V_U^* \otimes V_U \otimes V_U^* \otimes V_U}\right] = \frac{J}{4d^3 N}\sum_{i<j,ab}\left(T_{ia}^1 T_{jb}^1 + T_{ia}^2 T_{jb}^2 - T_{ia}^{\bar1*}T_{jb}^{\bar1*} - T_{ia}^{\bar2*}T_{jb}^{\bar2*}\right)^2 \tag{S.13}$$

Performing the average over the complete basis of Hermitian operators $T$, we have identities

$$\sum_a T_{ia}^1 T_{ia}^2 = d(X_{12})_i = d\sum_{a,b=0}^{d-1} \mathbb{1}_i \otimes (|a\rangle\langle b|)_i \otimes \mathbb{1}_i \otimes (|b\rangle\langle a|)_i$$

$$\sum_a T_{ia}^1 T_{ia}^{\bar2*} = d(P_{1\bar2})_i = d\left|I_i^{1\bar2}\right\rangle\left\langle I_i^{1\bar2}\right|, \quad \sum_a T_{ia}^1 T_{ia}^1 = d^2 \tag{S.14}$$

where $X_{12}$ swaps Hilbert space 1 and 2. We find $X_{12} = X_{21}$ and $X_{\bar1\bar2} = X_{\bar2\bar1}$, and when we act the swap operators on the states,

$$X_{12}\left|I^+\right\rangle = X_{21}\left|I^+\right\rangle = X_{\bar1\bar2}\left|I^+\right\rangle = X_{\bar2\bar1}\left|I^+\right\rangle = \left|I^-\right\rangle$$
$$X_{12}\left|I^-\right\rangle = X_{21}\left|I^-\right\rangle = X_{\bar1\bar2}\left|I^-\right\rangle = X_{\bar2\bar1}\left|I^-\right\rangle = \left|I^+\right\rangle \tag{S.15}$$

that is, the swap operators swap $|I^+\rangle$ with $|I^-\rangle$ and vice versa without a coefficient. $P_{1\bar2}$ is the projector onto Hilbert space 1 and $\bar2$, where

$$\left|I_i^{1\bar2}\right\rangle = \sum_{a=0}^{d-1} \mathbb{1} \otimes |a\rangle_i \otimes |a\rangle_i \otimes \mathbb{1} \tag{S.16}$$

with identity operators on Hilbert space $\bar1$ and 2. In general $(P_{a\bar b})_i = \left|I_i^{a\bar b}\right\rangle\left\langle I_i^{a\bar b}\right| = (P_{\bar b a})_i$ are defined similarly for $a, b = 1$ or 2. When we act the projectors on the states,

$$P_{\bar1 1}\left|I^-\right\rangle = P_{\bar2 2}\left|I^-\right\rangle = \left|I^+\right\rangle, \quad P_{\bar1 1}\left|I^+\right\rangle = P_{\bar2 2}\left|I^+\right\rangle = d\left|I^+\right\rangle \tag{S.17}$$
$$P_{1\bar2}\left|I^-\right\rangle = P_{\bar1 2}\left|I^-\right\rangle = d\left|I^-\right\rangle, \quad P_{1\bar2}\left|I^+\right\rangle = P_{\bar1 2}\left|I^+\right\rangle = \left|I^-\right\rangle \tag{S.18}$$

Namely $P_{\bar1 1}$ and $P_{\bar2 2}$ are "+projectors" that project onto the state $|I^+\rangle$, while $P_{1\bar2}$ and $P_{\bar1 2}$ are "-projectors" project onto the state $|I^-\rangle$.

We need one more ingredient, the qudit permutation symmetry, to simplify our calculations. If the initial state $\sigma(0)$ is invariant under arbitrary permutations of qudits, then $|\sigma(0)\otimes\sigma(0)\rangle\rangle$ is invariant under permutation of qudits

in the four-replica Hilbert space $\mathcal{H}^{(\bar{1})} \otimes \mathcal{H}^{(1)} \otimes \mathcal{H}^{(\bar{2})} \otimes \mathcal{H}^{(2)}$. After averaging, the time evolution process also has the qudit permutation symmetry, so we have

$$\dot{\mathcal{P}}_Q = \langle\!\langle X_Q | (-\mathcal{L}_U - \mathcal{L}_M) \overline{|\sigma(t) \otimes \sigma(t)\rangle\!\rangle} = \dot{\mathcal{P}}_n = \langle\!\langle X_n | (-\mathcal{L}_U - \mathcal{L}_M) \overline{|\sigma(t) \otimes \sigma(t)\rangle\!\rangle} \tag{S.19}$$

where we define $|X_n\rangle\!\rangle$ as the symmetrized $|X_Q\rangle\!\rangle$ for a subsystem $Q$ with $n$ qudits. Namely,

$$|X_n\rangle\!\rangle \equiv \frac{1}{\binom{N}{n}} \sum_{|Q|=n} |X_Q\rangle\!\rangle = \frac{1}{\binom{N}{n}} \sum_{|Q|=n} \left[ \bigotimes_{0 \leq i \leq N} |I_i^\pm\rangle \right] \tag{S.20}$$

where $|I_i^\pm\rangle = |I_i^-\rangle$ for $i \in Q$ and $|I_i^\pm\rangle = |I_i^+\rangle$ for $i \notin Q$, and the sum runs over all $\binom{N}{n}$ possible subsystems $Q$ with size $n$.

We write down the time evolution of the bra state $\langle\!\langle X_n|$ the Linbladian for the unitary part Eq.(S.13) using the identities Eq.(S.14, S.15, S.18) above.

$$\langle\!\langle X_n | (-\mathcal{L}_U) = -\frac{J}{4d^3 N} \left[ \frac{n(n-1)}{2} \left( 4d^2 \langle\!\langle X_{n-2}| - 4d^2 \langle\!\langle X_{n-2}| - 4d^4 \langle\!\langle X_n| + 4d^4 \langle\!\langle X_n| \right) \right.$$
$$+ n(N-n) \left( 4d^2 \langle\!\langle X_n| - 4d^3 \langle\!\langle X_{n-1}| - 4d^3 \langle\!\langle X_{n+1}| + 4d^4 \langle\!\langle X_n| \right)$$
$$\left. + \frac{(N-n)(N-n-1)}{2} \left( 4d^2 \langle\!\langle X_{n+2}| - 4d^4 \langle\!\langle X_n| - 4d^2 \langle\!\langle X_{n+2}| + 4d^4 \langle\!\langle X_n| \right) \right] \tag{S.21}$$
$$= J \frac{(N-n)n}{N} \left( \langle\!\langle X_{n-1}| + \langle\!\langle X_{n+1}| - \left( d + \frac{1}{d} \right) \langle\!\langle X_n| \right) \tag{S.22}$$

For clarity, in each of the three lines in Eq.(S.21), we write down four terms in the order that are the results of applying the swap, "+projector", "-projector", and identity operator on $\langle\!\langle X_n|$. The coefficient $\binom{n}{2}, \binom{n}{1}\binom{N-n}{1}, \binom{N}{2}$ in front of each of the three lines are the number of ways of choosing the qudits $i, j$ among the $n$ swap states $|I_i^-\rangle$ and $(N-n)$ maximally entangled states $|I_i^+\rangle$ in $\langle\!\langle X_n|$.

Now we move on to the measurement part $V_M$. Recall that we define the measurement rate $\lambda$ so that after each short time $\Delta t$, there is a small probability $p = N(d+1)\lambda\Delta t$ that one of the qudits, randomly chosen, is measured, and we do the average over the resulting states conditioned on only if we get identical measurement results in these two copies. So for a short time $\Delta t$, we have

$$\overline{|\sigma(t+\Delta t) \otimes \sigma(t+\Delta t)\rangle\!\rangle} = \overline{\mathbb{1}_{\mathcal{H}_S} \otimes V_M \sigma(t) V_M^\dagger \otimes \mathbb{1}_{\mathcal{H}_S} \otimes V_M \sigma(t) V_M^\dagger} |I^+\rangle_{1,\dots,N}$$
$$= \overline{V_M^* \otimes V_M \otimes V_M^* \otimes V_M} \, \overline{|\sigma(t) \otimes \sigma(t)\rangle\!\rangle}$$
$$= \left[ [1 - N\lambda(d+1)\Delta t] \mathbb{I} + \lambda(d+1)\Delta t \sum_i \left( |\psi_i^1\rangle \left|\psi_i^{\bar{1}}\right\rangle |\psi_i^2\rangle \left|\psi_i^{\bar{2}}\right\rangle \langle\varphi_i^1| \left\langle\varphi_i^{\bar{1}}\right| \langle\varphi_i^2| \left\langle\varphi_i^{\bar{2}}\right| \right) \right] \overline{|\sigma(t) \otimes \sigma(t)\rangle\!\rangle} \tag{S.23}$$
$$= \left[ 1 - \lambda(d+1)\Delta t \sum_i \left( \mathbb{I}_i - |\psi_i^1\rangle \left|\psi_i^{\bar{1}}\right\rangle |\psi_i^2\rangle \left|\psi_i^{\bar{2}}\right\rangle \langle\varphi_i^1| \left\langle\varphi_i^{\bar{1}}\right| \langle\varphi_i^2| \left\langle\varphi_i^{\bar{2}}\right| \right) \right] \overline{|\sigma(t) \otimes \sigma(t)\rangle\!\rangle} \tag{S.24}$$

and

$$\sum_i |\psi_i^1\rangle |\psi_i^{\bar{1}}\rangle |\psi_i^2\rangle |\psi_i^{\bar{2}}\rangle \langle\varphi_i^1| \langle\varphi_i^{\bar{1}}| \langle\varphi_i^2| \langle\varphi_i^{\bar{2}}|$$

$$= \sum_i \overline{U_i^* \otimes U_i \otimes U_i^* \otimes U_i} |0000\rangle_i \langle0000|_i \overline{W_i \otimes W_i^* \otimes W_i \otimes W_i^*}$$

$$= \sum_i \frac{1}{d^2-1} \left[ |I_i^+\rangle\langle I_i^+| + |I_i^-\rangle\langle I_i^-| - \frac{1}{d}\left(|I_i^+\rangle\langle I_i^-| + |I_i^-\rangle\langle I_i^+|\right) \right] |0000\rangle_i \langle0000|_i$$

$$\times \frac{1}{d^2-1} \left[ |I_i^+\rangle\langle I_i^+| + |I_i^-\rangle\langle I_i^-| - \frac{1}{d}\left(|I_i^+\rangle\langle I_i^-| + |I_i^-\rangle\langle I_i^+|\right) \right]$$

$$= \sum_i \frac{1}{d^2(d+1)^2} \left( |I_i^+\rangle\langle I_i^+| + |I_i^-\rangle\langle I_i^-| + |I_i^+\rangle\langle I_i^-| + |I_i^-\rangle\langle I_i^+| \right) \tag{S.25}$$

$$= \sum_i |S_i\rangle\langle S_i| \tag{S.26}$$

$$\text{with } |S_i\rangle = \frac{1}{d(d+1)}\left(|I_i^+\rangle + |I_i^-\rangle\right) \tag{S.27}$$

We have used the Weingarten function above, but in a similar spirit, we can also use Schur's lemma to get the same result. Then we can get the Linbladian for the measurement

$$\mathcal{L}_M = \lambda(d+1)\sum_i \left(\mathbb{I}_i - |S_i\rangle\langle S_i|\right), \tag{S.28}$$

From the definitions, we have $\langle I_i^\pm | I_i^\pm\rangle = d^2$ and $\langle I_i^\pm | I_i^\mp\rangle = d$ so

$$\langle\langle X_n|(-\mathcal{L}_M) = -\lambda(d+1)n\left(\langle\langle X_n| - \frac{1}{d(d+1)}\langle\langle X_{n-1}| - \frac{1}{d(d+1)}\langle\langle X_n|\right)$$

$$-\lambda(d+1)(N-n)\left(\langle\langle X_n| - \frac{1}{d(d+1)}\langle\langle X_n| - \frac{1}{d(d+1)}\langle\langle X_{n+1}|\right) \tag{S.29}$$

Finally, plugging $\mathcal{L}_U$ and $\mathcal{L}_M$ back to Eq. (S.9), we get the differential equation of the purity

$$\dot{\mathcal{P}}_n = \langle\langle X_n|(-\mathcal{L}_U - \mathcal{L}_M)\overline{|\sigma(t)\otimes\sigma(t)\rangle\rangle}$$

$$= J\frac{(N-n)n}{N}\left(\mathcal{P}_{n-1} + \mathcal{P}_{n+1} - \left(d+\frac{1}{d}\right)\mathcal{P}_n\right)$$

$$- \lambda n\left(\left(d+1-\frac{1}{d}\right)\mathcal{P}_n - \frac{1}{d}\mathcal{P}_{n-1}\right) - \lambda(N-n)\left(\left(d+1-\frac{1}{d}\right)\mathcal{P}_n - \frac{1}{d}\mathcal{P}_{n+1}\right)$$

$$= J\frac{(N-n)n}{N}\left(\mathcal{P}_{n-1} + \mathcal{P}_{n+1} - \left(d+\frac{1}{d}\right)\mathcal{P}_n\right) - \lambda N\left(d+1-\frac{1}{d}\right)\mathcal{P}_n + \frac{\lambda}{d}\left(n\mathcal{P}_{n-1} + (N-n)\mathcal{P}_{n+1}\right) \tag{S.30}$$

**Details of the Continuous (Large-N) Limit**

From Eq. (7) to Eq. (14): Following Eq. (7), we can write the purity differential equation in a matrix form, where we define a dimensionless variable $\alpha = \frac{\lambda}{dJ}$ as the rate of measurement over the rate of unitary evolution.

$$J^{-1}\dot{\mathcal{P}}_n = a_n\mathcal{P}_n + b_n\mathcal{P}_{n+1} + c_{n-1}\mathcal{P}_{n-1} \equiv \sum_m M_{nm}\mathcal{P}_m \tag{S.31}$$

$$c_{n-1} = \frac{n(N-n)}{N} + \alpha n, \; b_n = \frac{n(N-n)}{N} + \alpha(N-n), \; a_n = -\frac{n(N-n)}{N}\left(d+\frac{1}{d}\right) - \alpha d\left(d+1-\frac{1}{d}\right)N \tag{S.32}$$

The matrix is

$$M = \begin{pmatrix} a_0 & b_0 & 0 & \dots & & 0 \\ c_0 & a_1 & b_1 & \dots & & \\ 0 & c_1 & a_2 & \dots & & \\ & & \dots & & b_{N-1} \\ 0 & 0 & \dots & c_{N-1} & a_N \end{pmatrix} \tag{S.33}$$

The tridiagonal matrix $-M$ can be transformed into a symmetric (Hermitian) matrix, by a similarity transformation.

$$\mathcal{P}_n = \Lambda_n \phi_n, \quad -J M_{nm} \Lambda_m \Lambda_n^{-1} = H_{nm} \quad \Lambda_{n>0} \equiv \prod_{m=1}^{n} \sqrt{\frac{c_{m-1}}{b_{m-1}}}, \quad \Lambda_0 = 1 \tag{S.34}$$

$$H_{nn} = -J M_{nn} = -J a_n, \quad H_{n-1,n} = H_{n,n-1} = -J \sqrt{b_{n-1}c_{n-1}} \equiv -N\tau_n \tag{S.35}$$

Note the eigenvalues of $-M$ and $J^{-1}H$ are the same. The eigenvalue equation is $\sum_m H_{nm}\phi_m = E\phi_n$, with the minimum $E$ corresponding to the maximum eigenvalue of $M$. Since we are interested in the ratio $\mathcal{P}_n/\mathcal{P}_0$, the physically relevant energy eigenvalues are $E_a - E_0$ with $a > 0$.

We have the eigendecomposition of the Hermitian matrix

$$H = \sum_{a=0}^{N} \phi^a E_a (\phi^a)^\dagger, \quad M = -J^{-1}\Lambda H \Lambda^{-1} = -J^{-1}\Lambda \sum_{a=0}^{N} \phi^a E_a (\phi^a)^\dagger \Lambda^{-1} \tag{S.36}$$

where $\phi^a$ are the right eigenvectors. So

$$\mathcal{P}(t) = \Lambda \sum_a \phi^a e^{-E_a t} (\phi^a)^\dagger \Lambda^{-1} \mathcal{P}(0) = \Lambda \sum_a \left[ \sum_n (\phi_n^a)^* \mathcal{P}_n(0)\Lambda_n^{-1} \right] \phi^a e^{-E_a t} \tag{S.37}$$

$$\mathcal{P}_n(t) = \Lambda_n \sum_a \eta_a \phi_n^a e^{-E_a t}, \quad \text{where } \eta_a \equiv \sum_n (\phi_n^a)^* \mathcal{P}_n(0)\Lambda_n^{-1} \tag{S.38}$$

where the second line is written in the vector component form. Note if the lowest energy eigenvalue is unique,

$$\lim_{t\to\infty} \frac{\mathcal{P}_n(t)}{\mathcal{P}_0(t)} \simeq \lim_{t\to\infty} \frac{\Lambda_n \eta_0 \phi_n^0 e^{-E_0 t}}{\Lambda_0 \eta_0 \phi_0^0 e^{-E_0 t}} = \Lambda_n \frac{\phi_n^0}{\phi_0^0} \tag{S.39}$$

since $\Lambda_0 = 0$. Hence in infinite time the ratio $\mathcal{P}_n(t)/\mathcal{P}_0(t)$ is independent of the initial state $\mathcal{P}_n(0)$.

Following the main text, We use an ansatz $\phi_m = e^{-NA_m}$ to write the eigenvalue equation.

$$E = -J a_n - N\tau_n e^{-N(A_{n-1}-A_n)} - N\tau_{n+1} e^{-N(A_{n+1}-A_n)} \tag{S.40}$$

By taking the continuous limit, we can solve the problem analytically. Set $x \equiv \frac{n}{N}$, $\epsilon \equiv \frac{E}{N}$ and take $N \to +\infty$,

$$\epsilon = \frac{E}{N} = -J\frac{a_n}{N} - \tau(x) e^{\frac{A_n - A_{n-1}}{1/N}} - \tau(x+1/N) e^{-\frac{A_{n+1}-A_n}{1/N}} = V(x) - 2\tau(x)\left[\cosh\left(\partial_x A(x)\right) - 1\right] \tag{S.41}$$

with

$$J^{-1}V(x) = \lim_{N\to+\infty} \frac{-a_n}{N} - 2\tau_n = d\left(d+1-\frac{1}{d}\right)\alpha + \left(d+\frac{1}{d}\right)x(1-x) - 2\tau(x) \tag{S.42}$$

$$J^{-1}\tau(x) = \lim_{N\to+\infty} \frac{\sqrt{b_{n-1}c_{n-1}}}{N} = \sqrt{x(1-x)(1-x+\alpha)(x+\alpha)} \tag{S.43}$$

$$\Lambda(x) = \exp\left[\frac{N}{2}\int_0^x dy \log\frac{y(1-y)+\alpha y}{y(1-y)+\alpha(1-y)}\right] \tag{S.44}$$

Eq. (S.41) looks like the WKB equation for a quantum Hamiltonian in the classically forbidden region. Recall that for an action $A(x)$, the wave function looks like $\psi = \frac{1}{\sqrt{|\partial_x A(x)|}}e^{\frac{i}{\hbar}A(x)}$. So the momentum operator in the position basis $p = -i\hbar\partial_x$ becomes $p = \partial_x A(x)$ acting on the wave function. In this article the eigenvector $\phi(x)$ plays the role of the unnormalized wave function and the system size $N$ plays the role of $1/\hbar$. Since we divide everything by N as in $\epsilon = E/N$ in Eq. (S.41), we have the quantum Hamiltonian $\hat{h} = H/N$, with

$$\hat{h}(p,x) = -2\tau(x)\left[\cos(p) - 1\right] + V(x) \tag{S.45}$$

where $\cosh(p)$ is replaced with $\cos(p)$ in the classically forbidden region. Identify $\partial_t A = \epsilon$ so Eq.(S.41) can be viewed as the Hamiltonian-Jacobi equation.

To find the transition critical point, we observe that the $x = 1/2, p = 0$ point changes from a saddle point to a minimum at $\alpha = \frac{d-1}{2}$. We can check the second derivative along $x$ direction:

$$\frac{\partial^2}{\partial x^2} \hat{h}(0, x) = [V(x)]'' = 4J [\cosh \theta - \cosh \log d] \tag{S.46}$$

with $e^\theta = 2\alpha + 1$. The critical $\alpha$ is determined by $\theta = \log d$, which corresponds to $\alpha_c = \frac{d-1}{2}$. We can analytically solve for the two minima in the cusp phase or the minimum in the smooth phase.

$$J^{-1}V(x)_{\mathrm{min,cu}} = \alpha \left( d^2 + d - 1 - \frac{1}{d} - \frac{\alpha}{d} \right), \quad J^{-1}V(x)_{\mathrm{min,sm}} = \frac{1}{4} \left( d + \frac{1}{d} - 2 \right) + \alpha \left( d^2 + d - 2 \right) \tag{S.47}$$

at location $x_V$ (or $1 - x_V$), with

$$x_V = \frac{1}{2} \pm \sqrt{\frac{1}{4} - \frac{\alpha^2 + \alpha}{d^2 - 1}} \simeq \frac{1}{2} \pm \sqrt{\frac{d(\alpha_c - \alpha)}{d^2 - 1}} + O(\alpha_c - \alpha)^{\frac{3}{2}} \tag{S.48}$$

in the cusp phase and $x_V = \frac{1}{2}$ in the smooth phase. Eq.(S.47, S.48) have matched our results of finding the minimum of $V$ numerically.

The ground state energy is $\epsilon_0 \simeq \min_x V(x) + O\left(\frac{1}{N}\right)$. So as $N \to \infty$, we take $\epsilon = \epsilon_0 \equiv \min_x V(x)$. We can thus plug the solved $V(x)_{\mathrm{min}}$ back to find $A_{L,R}$ analytically. We rewrite

$$\epsilon = V(x) - 4\tau(x) \sinh^2 \left( \frac{1}{2} \partial_x A(x) \right) \tag{S.49}$$

$$\sinh \frac{\partial_x A}{2} = \pm \sqrt{\frac{V(x) - \epsilon_0}{4\tau(x)}} \tag{S.50}$$

Note that we need to pick the correct plus and minus signs so that after integration, the $A_{L,R}$ has a corresponding wave function $\phi^{L,R}(x) = e^{-NA_{L,R}(x)}$ localized in either the left or the right potential well respectively [Fig. 3(b)]. We find that

$$A_L(x) = A_L(0) + \int_0^1 2 \, \mathrm{sign}(x - x_V) \, \mathrm{arcsinh} \sqrt{\frac{V(x) - \epsilon_0}{4\tau(x)}} \tag{S.51}$$

and $A_R(x) = A_L(1 - x)$. Hence we can also get the entropy density $s(x) = A_{L,R}(x) + D(x) - A_{L,R}(0)$ with

$$D(x) \equiv -\frac{1}{N} \log \Lambda(x) = -\frac{1}{2} \int_0^x dy \log \frac{y(1 - y) + \alpha y}{y(1 - y) + \alpha(1 - y)} = -\frac{1}{2} \log \frac{x^x (1 - x)^{(1-x)} \alpha^\alpha (\alpha + 1)^{(\alpha+1)}}{(\alpha + 1 - x)^{(\alpha+1-x)} (\alpha + x)^{(\alpha+x)}} \tag{S.52}$$

comes from the similarity transformation. Note $D(x) = D(1 - x)$ is symmetric about $x = 1/2$. Our choice of $A_L$ or $A_R$ for the entropy density expression depends on the dominant solution that is switched at $x = 1/2$ for initially pure state or initially mixed state with $O(1)$ total entropy.

### Derivations of the Critical Behaviors

Having found the analytical solution of the entropy density $s$, we can solve for the system's critical behaviors. We want to calculate how the entropy density $s$ approaches its critical value at $\alpha = \alpha_c$, given a mixed initial state with one qudit of entropy. The density matrix $\sigma_N(0)$ at time $t = 0$ is $\sigma_N(0) = \frac{1}{d} \mathbb{I}_i \otimes_{j \neq i} |0\rangle \langle 0|$. The density matrix is then symmetrized with respect to all $N$ qudits for the ease of later calculations. The size-$n$ subsystem has a probability of $\frac{n}{N}$ to contain the one maximally mixed qudit, so the subsystem purity is

$$\mathcal{P}_n(0) = \mathrm{tr}_n(\sigma_n(0)^2) = \frac{n}{N} \frac{1}{d} + (1 - \frac{n}{N})1 = \frac{n + dN - dn}{dN} = x \frac{1 - d}{d} + 1 \tag{S.53}$$

with $x = n/N$. Recall from Eq. (18) that the total system entropy density at the long time is

$$S_n^{(2)}(t \to \infty) = -\log \frac{\mathcal{P}_n(t \to \infty)}{\mathcal{P}_0(t \to \infty)} = ND(x) - \log \frac{\eta_L \phi_L(x) + \eta_R \phi_R(x)}{\eta_L \phi_L(0) + \eta_R \phi_R(0)} \tag{S.54}$$

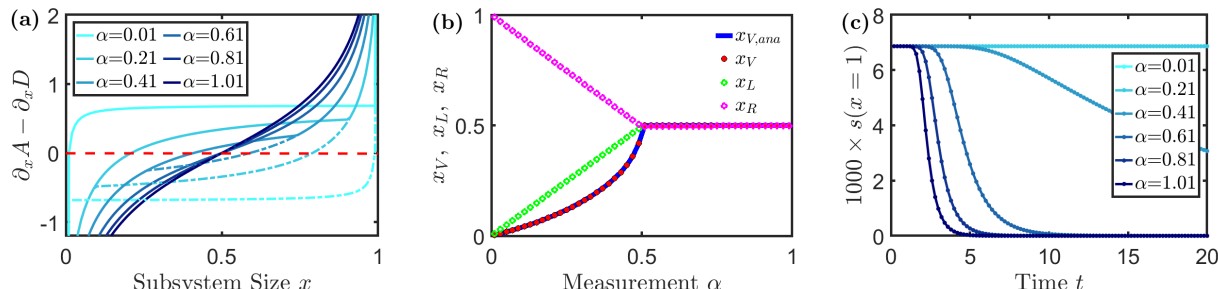

FIG. S.1: Steps to compute the total Entropy density $s(t \to \infty, x = 1)$. The calculation is done for $d = 2$ and $N = 101$. **(a)** Find the $x$ for the saddle point approximation Eq. (S.58). The dashed line around $y = 0$ is $\frac{1}{N}\partial_x \log \mathcal{P}(x, 0)$. For $\alpha < \alpha_c = 0.5$, $x_{L,R}$ are solutions of $\frac{1}{N}\partial_x \log \mathcal{P}(x, 0) = \partial_x A_{L,R}(x) - \partial_x D(x)$, where $\partial_x A_L(x) - \partial_x D(x)$ are the solid lines and $\partial_x A_R(x) - \partial_x D(x)$ are the dash-dotted lines. The solid lines and the dash-dotted lines split at $x_V$ and join at $1 - x_V$. For $\alpha > 0.5$, the $x_V$ is the solution of $\frac{1}{N}\partial_x \log \mathcal{P}(x, 0) = \partial_x A_0(x) - \partial_x D(x)$. In the limit $N \to \infty$, the dashed line is strictly at $y = 0$ and the whole graph is rotational symmetric. **(b)** $x_V, x_L, x_R$ extracted **(c)** Numerical simulation of the time evolution of the total entropy density. At the $O(N)$ long time the curves stabilize to a nonzero constant in the cusp phase or zero in the smooth phase.

In particular, since $D(1) = 0$, and $\phi_L(x) = \phi_R(1 - x)$, we have

$$S^{(2)}(x = 1, t \to \infty) = -\log \frac{\frac{\phi_R(0)}{\phi_L(0)} + \frac{\eta_R}{\eta_L}}{1 + \frac{\eta_R}{\eta_L}\frac{\phi_R(0)}{\phi_L(0)}} = -\log \frac{\eta_R}{\eta_L} \tag{S.55}$$

where in the last equality we have used that $\frac{\phi_R(0)}{\phi_L(0)} = \exp\{-N[A_R(0) - A_L(0)]\}$ is exponentially small for large $N$. In the large $N$ limit, we also have

$$\eta_L = \sum_{n=0}^{N} \phi_{L,n}^* \mathcal{P}_n(0)\Lambda_n^{-1} = N\int_0^1 dx \mathcal{P}(x, 0)e^{-N[A_L(x) - D(x)]}, \text{ and } \eta_R = N\int_0^1 dx \mathcal{P}(x, 0)e^{-N[A_R(x) - D(x)]} \tag{S.56}$$

We define

$$f_{L,R} \equiv -\frac{1}{N}\log \mathcal{P}(x, 0) + A_{L,R}(x) - D(x) \tag{S.57}$$

so the integral is dominated by the minimum of $f_{L,R}$. Set the minimum location of $f_{L,R}$ to be $x_{L,R}$. Then the saddle point approximation gives

$$\eta_{L,R} = N\int_0^1 dx \, \exp\left\{-N\left[f_{L,R}(x_{L,R}) + \frac{1}{2}f''(x_{L,R})(x - x_{L,R})^2 + \dots\right]\right\}$$

$$= Ne^{-Nf_{L,R}(x_{L,R})}\int_0^1 dx \, \exp\left\{-\frac{N}{2}f''(x_{L,R})(x - x_{L,R})^2 + \dots\right\}$$

$$\simeq Ne^{-Nf_{L,R}(x_{L,R})}\int_{-\infty}^{\infty} dx \, \exp\left\{-\frac{N}{2}f''(x_{L,R})(x - x_{L,R})^2\right\}$$

$$= \sqrt{\frac{2\pi N}{k_{L,R}}}\mathcal{P}(x_L, 0)\exp\{-N[A_{L,R}(x_{L,R}) - D(x_{L,R})]\} \tag{S.58}$$

For order $O(N^0)$ purity, or specifically $\mathcal{P}(x, 0) = x^{\frac{1-d}{d}} + 1$, we have $|\frac{1}{N}\log \mathcal{P}(x, 0)| \ll |A_{L,R}(x) - D(x)|$ so in the large $N$ limit the details of the initial state does not change the $x_L$ or $x_R$ and many calculations get simplified. In particular we have $k_L = k_R$, as well as $x_R = 1 - x_L$ since $A_R(x) = A_L(1 - x)$ by the definition of the left and right localized wave function. Plug these simplifications back we see that

$$\frac{\eta_R}{\eta_L} = \frac{\mathcal{P}(x_L, 0)}{\mathcal{P}(x_R, 0)} \tag{S.59}$$

Therefore, the long-time entropy at $x = 1$ is

$$S^{(2)}(1, \infty) = -\log \frac{\eta_R}{\eta_L} = -\log \frac{\mathcal{P}(x_R, 0)}{\mathcal{P}(x_L, 0)} = S^{(2)}(x_R, 0) - S^{(2)}(x_L, 0) \tag{S.60}$$

only depends on $x_L, x_R$ and the initial entropy density. We will find that for a mixed initial state with $O(1)$ entropy, finding $x_{L,R}$, the minima of functions $f_{L,R}$, is not too hard. We start from $0 = \partial_x f_{L,R}(x) = \partial_x \left[ -\frac{1}{N} \log \mathcal{P}(x, 0) - D(x) + A_{L,R}(x) \right]$. Note that

$$\partial_x \left[ -\frac{1}{N} \log \mathcal{P}(x, 0) \right] = -\frac{1}{N\mathcal{P}(x,0)} \frac{1-d}{d} = \frac{d-1}{N[x(1-d)+d]} \xrightarrow{N\to\infty} 0 \tag{S.61}$$

since $\mathcal{P}(x, 0)$ is not zero for $x \in [0, 1]$. So we want to solve for

$$0 = \partial_x A_{L,R}(x) - \partial_x D(x) = 2 \operatorname{arcsinh} \sqrt{\frac{V(x) - \epsilon_0}{4\tau(x)}} + \frac{1}{2} \log \frac{x(1-x) + \alpha x}{x(1-x) + \alpha(1-x)} \tag{S.62}$$

where the analytical expression of $\epsilon_0 = V(x)_{\min, cu}$ in the cusp phase is used. We will eventually find

$$x_L = \frac{\alpha}{d-1} \simeq \frac{1}{d-1} (\alpha - \alpha_c), \text{ and } x_R = 1 - \frac{\alpha}{d-1} \simeq 1 - \frac{1}{d-1} (\alpha - \alpha_c) \tag{S.63}$$

for $\alpha < \alpha_c = \frac{d-1}{2}$ in the cusp phase. So

$$S^{(2)}(1, \infty) = S^{(2)}(x_R, 0) - S^{(2)}(x_L, 0) = -\log \left( \frac{\alpha + 1 - d}{d} + 1 \right) + \log \left( \frac{-\alpha}{d} + 1 \right) = -\log \left( \frac{1+\alpha}{d-\alpha} \right) \tag{S.64}$$

Setting $w = \alpha - \alpha_c$ we have

$$S^{(2)}(1, \infty) = -\log \left( \frac{1 + \frac{2w}{d+1}}{1 - \frac{2w}{d+1}} \right) \simeq - \left[ \frac{4w}{d+1} + O(w^3) \right] \tag{S.65}$$

in the cusp phase. In the smooth phase $x_L = x_R$ so $S^{(2)}(1, \infty) = 0$.

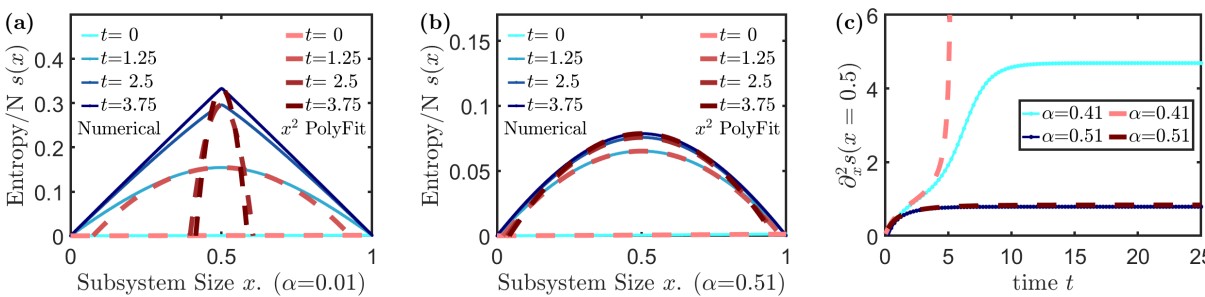

FIG. S.2: Second derivative of the entropy density at the turning point in finite time, with an initially pure state. See Eq.(S.78, S.81). The calculation is done for $d = 2$ and $\alpha_c = 0.5$ **(a)(b)** The time evolution of the subsystem entropy density $s(t, x)$ at (a) $\alpha = 0.01$ in the cusp phase and at (b) $\alpha = 0.51$ in the smooth phase. The second derivative is extracted from a second-order polynomial fit with 10 data points around the turning point. **(c)** The comparison between the extracted second derivatives and the analytical ones for $\alpha = 0.41$ in the cusp phase and $\alpha = 0.51$ in the smooth phase. As $N \to \infty$, the cusp phase curve will go to infinity at a finite time as expected.

For a pure initial state or an initial state with only $O(1)$ total entropy, we need to switch the dominant solution at $x = 1/2$. It is thus interesting to calculate the first derivative and the second derivative of entropy density at $x = 1/2$. Using Eq. (S.50) and Eq. (S.52), we find that

$$\partial_x s|_{x=1/2} = \partial_x A|_{x=1/2} + \partial_x D|_{x=1/2} = \pm 2 \operatorname{arcsinh} \sqrt{\frac{V(x) - \epsilon_0}{4\tau(x)}} \Bigg|_{x=1/2} + \log(1)$$

$$= \pm 2 \operatorname{arcsinh} \sqrt{\frac{\frac{1}{4}\left(d + \frac{1}{d}\right) - \left(\frac{1}{2} + \alpha\right) + \frac{1}{d}(\alpha^2 + \alpha)}{1 + 2\alpha}} + 0 = \pm \log \left( \frac{d}{1 + 2\alpha} \right) \tag{S.66}$$

in the cusp phase and 0 in the smooth phase. The $\pm$ sign was due to the switch of dominant solution at, for example, $x = 1/2$ given a pure initial state. For $x < 1/2$, we have

$$\partial_x s|_{x=1/2} = -\log\left(1 + \frac{2}{d}(\alpha - \alpha_c)\right) \simeq \frac{2}{d}(\alpha_c - \alpha) + O(\alpha_c - \alpha)^2 \tag{S.67}$$

For the second derivative, we find

$$\partial_x^2 s|_{x=1/2} = \partial_x^2 A|_{x=1/2} + \partial_x^2 D|_{x=1/2} = \frac{2\sqrt{(2\alpha - d + 1)(2d\alpha + d - 1)}}{(1 + 2\alpha)\sqrt{d}} - \frac{4\alpha}{1 + 2\alpha} \simeq \left(-2 + \frac{2}{d}\right) + O(\alpha - \alpha_c)^{\frac{1}{2}} \tag{S.68}$$

in the smooth phase. In the cusp phase, we find that $\partial_x^2 A|_{x=1/2}$ is infinite as expected. For the case that the switch of sign happens at $x > 1/2$ (e.g. an initially mixed state with $O(N)$ total entropy), we have

$$\partial_x^2 A_L|_{x=1/2} + \partial_x^2 D|_{x=1/2} = 0 - \frac{4\alpha}{1 + 2\alpha} = -\frac{4\alpha}{1 + 2\alpha} \simeq \left(-2 + \frac{2}{d}\right) + O(\alpha - \alpha_c) \tag{S.69}$$

where we can check that $\partial_x^2 A_L|_{x=1/2} + \partial_x^2 D|_{x=1/2}$ is zero at $\alpha = 0$ as expected.

We will also derive the time-dependent formalism in detail here. We copy from Eq. (25) and Eq. (S.41),

$$\partial_t A(x,t) = V(x) + 2\tau(x) - 2\tau(x)\cosh(\partial_x A(x,t)) \tag{S.70}$$

We expand around $x = 1/2$ and take use of the numerical observation that $\partial_x A(x,t) = 0$ at $x = 1/2$ and early time for both phases. We first take a spatial derivative of Eq. (S.70), and set $f(x,t) = \partial_x A(x,t)$ at early time,

$$\partial_t \partial_x A(x,t) = \partial_x (V + 2\tau)(x) - 2\partial_x \tau(x)\cosh(\partial_x A(x,t)) - 2\tau(x)\sinh(\partial_x A(x,t))\partial_x^2 A(x,t) \tag{S.71}$$

$$\partial_t f(x,t) = \partial_x (V + 2\tau)(x) + 2\partial_x \tau(x) - 2\partial_x \tau(x)\cosh(f(x,t)) - 2\tau(x)\sinh(f(x,t))\partial_x f(x,t) \tag{S.72}$$

and Taylor-expand around $x = 1/2$ to get

$$f(x) = 0 + (x - 1/2)f'(1/2) + O((x - 1/2)^2) = -z(u - \tilde{u}) + O(z^2) \tag{S.73}$$

where we have set $z = x - 1/2$, $u = -\partial_x^2 s(x)|_{x=1/2}$, $\tilde{u} = -\partial_x^2 D(x)|_{x=1/2} = \frac{4\alpha}{1+2\alpha}$ so that $u - \tilde{u} = -f'(1/2) = -\partial_x^2 A(x,t)|_{x=1/2}$. Note that $u(t = 0) = 0$ is our initial condition and $\partial_t \tilde{u} = 0$. Similarly,

$$(V + 2\tau)(x) = (V + 2\tau)(1/2) + z(V + 2\tau)'(1/2) + \frac{z^2}{2}(V + 2\tau)''(1/2) + O(z^3) = J\left[\left(d + 1 - \frac{1}{d}\right)\alpha + 0 - z^2\left(d + \frac{1}{d}\right)\right] \tag{S.74}$$

$$\tau(x) = \tau(1/2) + z\tau'(1/2) + \frac{z^2}{2}\tau''(1/2) + O(z^3) = J\left[\frac{1 + 2\alpha}{4} - \frac{z^2}{2}\left(1 + 2\alpha + \frac{1}{1 + 2\alpha}\right)\right] \tag{S.75}$$

We substitute them back to Eq. (S.72) and keep the lowest order of $z$,

$$z\partial_t u(t) = -\partial_x (V + 2\tau)(x) + 2\partial_x \tau(x)\cosh\left[-z(u - \tilde{u})\right] + 2\tau(x)\sinh\left[-z(u - \tilde{u})\right]\left[-(u - \tilde{u})\right] \tag{S.76}$$

$$J^{-1}\partial_t u(t) = 2\left(d + \frac{1}{d}\right) - 2\left(1 + 2\alpha + \frac{1}{1 + 2\alpha}\right) + \frac{1 + 2\alpha}{2}(u - \tilde{u})^2 \equiv \tilde{a}(u - \tilde{u})^2 + \tilde{b} \tag{S.77}$$

There is a critical point at $\tilde{b} = 0$. For $\tilde{b} \geq 0$, $\dot{u}$ is always positive, so that the second derivative increases without bound, indicating a Page curve with a discontinuity. For $\tilde{b} < 0$, $u$ will saturate at the long time. The critical point is at $\alpha_c = \frac{d-1}{2}$. For $\alpha < \alpha_c$, the solution of the differential equation is

$$u = \sqrt{\frac{\tilde{b}}{\tilde{a}}}\tan\left(J\sqrt{\tilde{a}\tilde{b}}(t - t_0)\right) + \tilde{u} \tag{S.78}$$

which diverges at a finite time $t_c$ with

$$t_c - t_0 = \frac{\pi}{2J\sqrt{\tilde{a}\tilde{b}}} = \frac{\pi\sqrt{d}}{2J\sqrt{(d - 1 - 2\alpha)(2d\alpha + d - 1)}} \simeq \frac{\pi}{J\sqrt{8\left(d - \frac{1}{d}\right)(\alpha_c - \alpha)}} + O(\alpha_c - \alpha)^{\frac{1}{2}} \tag{S.79}$$

and

$$t_0 = \frac{\arctan\left[\sqrt{\frac{\tilde{a}}{\tilde{b}}}\tilde{u}\right]}{J\sqrt{\tilde{a}\tilde{b}}} = \frac{\arctan\left[\frac{2\alpha}{\sqrt{\tilde{a}\tilde{b}}}\right]}{J\sqrt{\tilde{a}\tilde{b}}} \simeq \frac{\frac{\pi}{2} - \frac{\sqrt{\tilde{a}\tilde{b}}}{2\alpha} + O(\alpha_c - \alpha)^{\frac{3}{2}}}{J\sqrt{\tilde{a}\tilde{b}}} \simeq \frac{\pi}{J\sqrt{8\left(d - \frac{1}{d}\right)(\alpha_c - \alpha)}} + O(1) \tag{S.80}$$

For $\alpha > \alpha_c$, the solution is

$$u = -\sqrt{\frac{|\tilde{b}|}{\tilde{a}}} \coth\left(J\sqrt{\tilde{a}|\tilde{b}|}(t - t_0)\right) + \tilde{u} \tag{S.81}$$

we have

$$u(t \to \infty) = -\sqrt{\frac{|\tilde{b}|}{\tilde{a}}} + \tilde{u} \simeq \left(2 - \frac{2}{d}\right) - \sqrt{\frac{8(d^2 - 1)(\alpha - \alpha_c)}{d^3}} + \frac{4(\alpha - \alpha_c)}{d^2} + O(\alpha_c - \alpha)^{\frac{3}{2}} \tag{S.82}$$

which matches $-\partial_x^2 s|_{x=1/2}$ in Eq. (S.68).

$$t_0 = \frac{-\operatorname{arccoth}\left[\sqrt{\frac{\tilde{a}}{|\tilde{b}|}}\tilde{u}\right]}{J\sqrt{\tilde{a}|\tilde{b}|}} \simeq \frac{1}{1 - d} + O(\alpha_c - \alpha) \tag{S.83}$$

And for $t > t_c = t_0 + \frac{\pi}{2J\sqrt{\tilde{a}|\tilde{b}|}}$, we have

$$u(t \to \infty) - u < \left(-1 + \coth\frac{\pi}{2}\right)\sqrt{\frac{|\tilde{b}|}{\tilde{a}}} \simeq 0.09\sqrt{\frac{8(d^2 - 1)}{d^3}}\sqrt{\alpha - \alpha_c} + O(\alpha - \alpha_c)^{\frac{3}{2}} \tag{S.84}$$

Both solutions match our numerical simulations in Fig S.1 (c).

## The Relation Between the Averaged Purity with Averaged Von Neumann Entropy

The following discussion is quite similar to that about random tensor networks in appendix A of Ref. [23]. Let's denote the ensemble of states at time $t$ obtained by the random measurement as $\rho_\zeta(t)$. For simplicity we denote all random parameters, including $J(t), \psi(t), \phi(t)$ and the sites being measured, as $\zeta$. We define $\rho_\zeta(t)$ as a normalized state, and denote the corresponding un-normalized state as $\pi_\zeta(t)$. $\pi_\zeta(t)$ is linear in the initial state $\rho_{\text{in}}$. The probability of $\rho_\zeta(t)$ is $p_\zeta(t) = \operatorname{tr}(\pi_\zeta(t))$. The purity we studied is defined by

$$e^{-S_R^{(2)}} = \frac{\int d\zeta \operatorname{tr}\left(X_R \pi_\zeta(t)^{\otimes 2}\right)}{\int d\zeta p_\zeta(t)^2} \tag{S.85}$$

Now we consider the averaged von Neumann entropy of this ensemble

$$\overline{S_R} = -\int d\zeta p_\zeta(t) \operatorname{tr}\left(\rho_{\zeta R}(t) \log \rho_{\zeta R}(t)\right) \tag{S.86}$$

Using the standard replica trick,

$$\begin{aligned}
\overline{S_R} &= -\int d\zeta p_\zeta(t) \frac{\partial}{\partial n} \log \operatorname{tr}\left(\rho_{\zeta R}(t)^n\right)\Big|_{n \to 1} \\
&= -\frac{\partial}{\partial n}\int d\zeta p_\zeta(t)\left(\log \operatorname{tr}\left(\pi_{\zeta R}(t)^n\right) - \log p_\zeta(t)^n\right)\Big|_{n \to 1} \\
&= -\frac{\partial}{\partial n}\int d\zeta\left[\operatorname{tr}\left(\pi_{\zeta R}(t)^n\right) - p_\zeta(t)^n\right]\Big|_{n \to 1}
\end{aligned} \tag{S.87}$$

In addition if we use the fact that

$$\int d\zeta \, \text{tr}\left(\pi_{\zeta R}(t)\right) = \int d\zeta \, p_\zeta(t) = 1 \tag{S.88}$$

we can write

$$\overline{S_R} = -\left.\frac{\partial}{\partial n} \frac{Z_{Rn}}{Z_{\emptyset n}}\right|_{n \to 1} \tag{S.89}$$

$$Z_{Rn} = \int d\zeta \, \text{tr}\left(\pi_{\zeta R}(t)^n\right) \tag{S.90}$$

$$Z_{\emptyset n} = \int d\zeta \, p_\zeta(t)^n \tag{S.91}$$

Compare this with Eq. (3) we see that $\mathcal{P}_Q = Z_{R2}$, $\mathcal{P}_\emptyset = Z_{\emptyset 2}$. Thus if we generalize the quantity we computed in Eq. (3) to

$$e^{-(n-1)S_R^{(n)}} = \frac{Z_{Rn}}{Z_{\emptyset n}} \tag{S.92}$$

then $S_R^{(n)} \to \overline{S_R}$ for $n \to 1$, even if for integer $n > 1$ it is not an average of Renyi entropy.