# Peer review of "Measurement-Induced Entanglement Phase Transition in Random Bilocal Circuits"

_SciPost Physics_

## Round 2 · Referee Report · Yi-Zhuang You (Referee 1) · 2025-2-28

Strengths

  1. The paper studies a simple yet insightful N-qudit Brownian circuit model with random interactions and measurements, which exhibits a measurement-induced entanglement phase transition
  2. The model can be mapped to a one-dimensional quantum chain in the semi-classical limit, allowing for analytical study of the critical behaviors and properties of the model
  3. The authors show that the model has two distinct phases in the long-time behavior of the total system entropy and subsystem entropy, with the low measurement rate phase exhibiting a "Page curve"-like first-derivative discontinuity
  4. The analytical approach and the mapping to a 1D quantum chain enable a detailed understanding of the critical properties of the measurement-induced phase transition in this non-local model, going beyond previous works that relied more on phenomenological effective field theory and numerical results

Report

This paper presents a novel and insightful study of the measurement-induced entanglement phase transition in a simple yet non-trivial N-qudit Brownian circuit model with random interactions and measurements. The authors show that despite the model's lack of spatial locality, its subsystem purities can be mapped to a one-dimensional quantum problem, allowing for detailed analytical study of the measurement-induced entanglement transition.

Overall, this paper presents a solid and insightful study of measurement-induced entanglement phase transitions in a non-local model. The analytical tractability of the model and the rich critical behaviors uncovered make this work a valuable contribution to the field. It is suprising to see that the behavior of this (0+1)d simple all-to-all interacting system mimics many important aspects of entanglement transition in higher dimensions. In my view, this paper meets the high standards of SciPost and I would recommend its publication.

Requested changes

  1. It is a bit unintuitive how to relate shape of ground-state wavefunction to the shape of the entropy function. It would be better if the authors can provide plots about the functions A(x) and D(x) also at alpha=0.31, such that the reader can get a step by step visualization of how the two peak vs single peak behaviors of the wave function gets translated into the no cusp vs cusp behaviors of the entanglement entropy.

Recommendation

Publish (surpasses expectations and criteria for this Journal; among top 10%)

---

## Round 2 · Referee Report · Anonymous (Referee 2) · 2025-3-24

Report

The manuscript "Measurement-Induced Entanglement Phase Transition in Random Bilocal Circuits" considers a monitored N-qudit model with all-to-all interactions. Such models are often found to exhibit a measurement-induced entanglement transition as the measurements may impede the entanglement growth. The authors are able to map the model to a one-dimensional quantum chain, which allows for analytical and numerical results, and show that there is a transition between a phase with a cusp in the entropy curve and a phase with a smooth entropy curve.

I find the analytical approach used in this work and the results to be certainly interesting, and to fall within the high standards of this journal. However, the manuscript, as it is currently presented, lacks motivation and physical insights, as well as making a clear case for the significance of the results (see specifically the detailed requests for revisions in Comment 1). I believe that if the authors address these issues, the manuscript will be suitable for publication in SciPost Physics.

Requested changes

1) I find there is a significant lack of motivation throughout the manuscript, as well as a need to spell out the significance of the results. In general, I believe the authors should think about how to interest the reader and highlight their results properly. Below I list more specific changes, which should significantly improve the manuscript.

1a) In the abstract, after listing the results, I feel there should be one or two sentences saying why these results are interesting, and what this work implies for future research.

1b) In the introduction, there seems to be a lack of motivation. What is the question that has not been addressed before, that the authors want to answer? How do the new results go beyond what has been done before? (E.g. one could argue that the purification picture is enough, as the main text identifies the transition in Fig. 2a with the purification transition.) At the end of the introduction, there should again be some sentence giving the reader an idea of the significance of these results.

1c) I often find that sections lack a synthesis paragraph at the end, which briefly summarises what the authors achieved - this makes the reader question the significance. The worst example of that is the "Critical behavior" section, which ends with an equation without any comment as to why the reader should find this interesting. Hence, I would like to ask authors to add brief summary paragraphs at the end of the "Purity evolution", "Large-N limit", and "Critical behavior" sections, as I believe these would make it clear to the reader what has been done, include the physical insights, but also highlight the significance of the results.

1d) In the discussion section, the authors summarise their results in one sentence. I find this lacks proper synthesis and again does not highlight the results appropriately. For example, clearly, the authors are able to identify phases that mimic the volume and the area law phases, i.e. the "cusp phase" and the "smooth phase", and this should be commented on. Comments on their analytical methodology are also needed.

1e) Again in the discussion, I would urge the authors to show how these results are interesting for the broader audience, that goes beyond the obvious (measurement-induced transitions / monitored quantum systems). Which fields or research will be impacted by this paper? I presume, for example, these results would be interesting for people exploring black hole entropy dynamics?

2) The authors write "Two qudits are randomly chosen and coupled by a two-qudit gate with a certain probability. Quantum measurement is randomly applied to one of these qudits with a certain rate." Is the measurement applied to any of the N qudits or is it applied to one of the two qudits that were coupled by a two-qudit gate? The sentences as written imply the second scenario, but the discussion below Eq. 2 implies the first scenario. The authors should make sure that this is precisely defined.

3) Is there a particular reason for making N odd in Fig. 2a? Also, what is N equal to in Fig. 2b?

4) The authors should also consider updating their bibliography - their preprint is dated 3 years ago and since then many papers on monitored all-to-all models have been published. Hence, it may also be necessary to update the introduction slightly.

5) Equations should be properly punctuated, otherwise the text is hard to read.

6) Minor grammar corrections: * "In general, time evolution of" -> "In general, the time evolution of" * "an mixed initial state" -> "a mixed initial state" * "For simplicity we denote" -> "For simplicity, we denote" * "with details reserved to the supplemental materials" -> "with details reserved for the supplemental material" * "as ratio of the rate" -> "as the ratio of the rate" * "A generalization to Brownian circuit" -> "A generalization to the Brownian circuit" * "The dashed lines are analytic results on the contribution" -> "The dashed lines are analytic results of the contribution" * "The vector $\phi_n$ satisfy" -> "The vector $\phi_n$ satisfies" * "the analog of Hamilton-Jacobi equation" -> "the analog of the Hamilton-Jacobi equation" * "In the following we will analyze" -> "In the following, we will analyze" * "Similarly we found" -> "Similarly, we found" * "If the initially state is a pure state" -> "If the initial state is a pure state"

Recommendation

Ask for major revision

---

## Editorial Decision

awaiting_resubmission